# Transmission of *Escherichia coli* from Manure to Root Zones of Field-Grown Lettuce and Leek Plants

**DOI:** 10.3390/microorganisms9112289

**Published:** 2021-11-03

**Authors:** Leo van Overbeek, Marie Duhamel, Stefan Aanstoot, Carin Lombaers van der Plas, Els Nijhuis, Leo Poleij, Lina Russ, Patricia van der Zouwen, Beatriz Andreo-Jimenez

**Affiliations:** Wageningen Plant Research, Wageningen University and Research (WUR), 6708 PB Wageningen, The Netherlands; marie.duhamel@wur.nl (M.D.); stefan.aanstoot@wur.nl (S.A.); carin.lombaers@wur.nl (C.L.v.d.P.); els.nijhuis@wur.nl (E.N.); leo.poleij@wur.nl (L.P.); lina.russ@wur.nl (L.R.); patricia.vanderzouwen@wur.nl (P.v.d.Z.); beatriz.andreojimenez@wur.nl (B.A.-J.)

**Keywords:** *Escherichia coli*, plants, manure, transmission, food safety, metagenome, polyphasic detection

## Abstract

Pathogenic *Escherichia coli* strains are responsible for food-borne disease outbreaks upon consumption of fresh vegetables and fruits. The aim of this study was to establish the transmission route of *E. coli* strain 0611, as proxy for human pathogenic *E. coli*, via manure, soil and plant root zones to the above-soil plant compartments. The ecological behavior of the introduced strain was established by making use of a combination of cultivation-based and molecular targeted and untargeted approaches. Strain 0611 CFUs and specific molecular targets were detected in the root zones of lettuce and leek plants, even up to 272 days after planting in the case of leek plants. However, no strain 0611 colonies were detected in leek leaves, and only in one occasion a single colony was found in lettuce leaves. Therefore, it was concluded that transmission of *E. coli* via manure is not the principal contamination route to the edible parts of both plant species grown under field conditions in this study. Strain 0611 was shown to accumulate in root zones of both species and metagenomic reads of this strain were retrieved from the lettuce rhizosphere soil metagenome library at a level of Log 4.11 CFU per g dry soil.

## 1. Introduction

Pathogenic *Escherichia coli* strains are responsible for food-borne diseases upon consumption of fresh vegetables and fruits. In a survey involving prevalence of food-borne pathogens in freshly-consumed products, performed in the Netherlands, pathogenic *E. coli* O157:H7 was found present in 0.11% of the investigated cases [1]. The largest food-borne outbreak related to consumption of plant-derived and freshly consumed products was in the Hamburg area (Germany) in May 2011. Contamination of fenugreek sprouts by enterohemorrhagic *E. coli* (EHEC) O104:H4 resulted in this particular case in 3842 infected individuals, of which 830 showed symptoms of hemolytic uremic syndrome (HUS) and 46 deceased [2,3,4]. The strain responsible for the outbreak was an enteroaggregative *E. coli* strain and its efficient adhesion to epithelial cells in human gut systems was the most likely reason for the high number of HUS incidences [2,3]. Contamination of the fenugreek sprouts must have occurred via seed contamination, as evidenced by epidemiological data, although the pathogen could not be traced back from the original seed batches [2]. How fenugreek seeds became contaminated with the pathogen remained unclear, and therefore, more ecological data on transmission of human pathogens to growing plants are needed.

*E. coli* can enter soil via manure application [5,6,7] and irrigation [8,9], and therefore, manure and water are considered to be important transmission vehicles for transfer of *E. coli* to plant root zones. High incidences of *E. coli* in open water sources, used for irrigation, often result from fecal contaminations, and therefore, animal feces must be considered as primary sources of both water and soil contamination by *E. coli*. An *E. coli* O157:H7 prevalence of 15.4% among bovine manure samples was found in a multi-county survey done in California, with density levels ranging between Log 3.1–5.3 colony forming units (CFU) per g manure [10]. The intestinal track system of endotherms was considered to be the primary habitat for *E. coli*, whereas natural environmental ecosystems often serve as secondary habitats for this species [9]. On pasture land grazed by heifers, an STEC population was shown to circulate between grass and the grazing population [11], indicating that microbial exchanges must have taken place between grazing herbivores and the plant cover. Ecosystems are interconnected by cycling of microbial subpopulations from the environment, in particular from soil, to plants and animals, and finally, these micro-organisms end up in food production chains [12].

Pathogenic *E. coli* can live in association with plants. Upon inoculation, *E. coli* O157:H7 was isolated from surface-sterilized spinach plants [13]. However, internal plant colonization only sporadically occurred, as was demonstrated for *E. coli* O157:H7 in spinach and lettuce plants [14] and for a Shigatoxin-producing *E. coli* (STEC) strain, an extended-spectrum β-lactamase-producing *E. coli* (ESBL) strain, and an enteroaggregative *E. coli* strain in pea plants [15]. Furthermore, *E. coli* was found to persist on seeds and shown to grow out with seedling growth [15,16]. On lettuce seeds, *E. coli* O157:H7 was demonstrated to survive for two years and still able to grow out with the young plants emerging from the seeds [16] and three different *E. coli* strains were able to colonize young plants from seeds [15]. Cells of *E. coli* also were shown to adhere rapidly to leaf surfaces, and once adhered, they showed a remarkable resistance to removal by disinfecting agents [17,18]. The combination of colonization and adherence associated genes were demonstrated to play central roles in plant adherence by *E. coli* O157:H7 [19]. Factors related to leaf axis (adaxial or abaxial sides), age and surface topography as well as local environmental circumstances, such as nutrient availability and temperature, were shown to play decisive roles in plant colonization by *E. coli* [18]. The root zone of plants is the habitat that is commonly favored by *E. coli*, as demonstrated for cabbage [20], lamb’s lettuce [19], lettuce [5,6,21,22] pea [15], radish [5] and grass plants [23]. *Escherichia coli* can accumulate on/inside roots and in rhizosphere soil (together named the ‘plant root zone’) when plants come into contact with contaminated soil. It is a key question if plant root zones play pivotal roles in further spread of *E. coli* in the above-soil, and often edible plant compartments.

Soil and plant roots are relevant habitats in food production systems, where *E. coli* can survive over longer periods in time. *Escherichia coli* remained viable for 85 days in tropical soil [20], whereas in another study, *E. coli* survived for over four months in soil and coverage by vegetation was shown to play an important role in survivability [24]. Presence of *E. coli* was confirmed in a soil that had received manure nine years before [25], suggesting that *E. coli* may become an intrinsic component of soil microbial communities upon introduction via manure into soils. An increase in soil temperature and amplitude in daily temperature oscillation [26], higher pH [27] and organic carbon content [28] resulted in extended survival of *E. coli* in bulk soil. Differences in bacterial diversity and evenness in 36 different Dutch soils did not significantly impact the survival time of an introduced *E. coli* O157:H7 strain, but a higher soil bacterial diversity resulted in significant higher irregularity, which is a measure for predictability in *E. coli* O157:H7 decline in soil [29]. A negative correlation was found between soil microbial diversity and *E. coli* O157:H7 survival and the introduced strain was shown to be less competitive in both the amount of nutrient resource usage and consumption rate in comparison with 40 selected soil bacterial strains [30]. Competition for nutrient resources was also found to be the main factor in *E. coli* strains with Shigatoxin (*stx*) genes (genes encoding for toxins associated with enterohemorrhagic diseases) in their genomes [27]. In this study, it was shown that strains without *stx* genes survived longer in bulk soil than the ones carrying these genes. However, the roles that *stx* genes played in *E. coli* survival in the environment seems controversial, as no difference in pea root colonization was observed with three distinct *E. coli* strains, of which one contained *stx* genes [15].

The aim of this study was to establish the transmission route of an introduced *E. coli* strain, as proxy for a human pathogenic bacterium, via manure and soil to the root zones and the above-ground compartments of two distinct plant species (monocot and a dicot species), cultivated in the open field. Because the ecological behavior of the introduced strain was expected to be determined by the bacterial community composition near the roots, the impact of manure application with and without introduced *E. coli* strain on the bacterial community composition of the rhizosphere soils of both plant species was assessed. Detection of human pathogens in plant production systems is important, and therefore, the introduced *E. coli* strain was traced back via a polyphasic approach. It was anticipated that the use of a polyphasic approach would allow us to compare quantitative data with data obtained via untargeted, DNA sequencing-based approaches to evaluate the usefulness of targeted and untargeted approaches in screening for presumed pathogenic species in plant-soil microbiomes.

## 2. Materials and methods

### 2.1. E. coli Strain 0611 Cultivation, CFU Recovery and DNA Extraction

*Escherichia coli* strain 0611 was isolated from morning glory leaves from Thailand, imported as a culinary herb into the Netherlands [31]. The strain was selected from the leaves on the basis of an intrinsic resistance towards cefotaxime (Cf). Cells of strain 0611 were standard grown overnight from frozen stock solutions (stored at −70 °C in 20% glycerol) under shaking (180 RPM) at 37 °C in Luria Bertani broth (LB: Tryptone, 10 g; yeast extract, 5 g; NaCl, 10 g; dissolved in one L water and autoclaved for 20 min at 121 °C) amended with 0.1% Cf. For inoculation into manure and for DNA extraction, cells from overnight cultures were collected by centrifugation at 7000× *g* for 5 min and resulting cell pellets were suspended in Ringer’s solution (¼ strength Ringer, OXOID BR0052, Basingstoke, UK; one tablet dissolved in 0.5 L demineralized water followed by autoclaving). For recovery of strain 0611 CFUs from manure, bulk and rhizosphere soils, roots and leaves, brilliance *E. coli*/coliform selective agar medium (BECSA, OXOID CM1046, Basingstoke, UK), amended with 0.1% Cf, was used. For whole genome sequencing of strain 0611, the same DNA extraction, DNA sequencing and sequence analysis procedures were used as described before [23].

### 2.2. Soil Treatments and Experimental Field Design

A field study was conducted at the Unifarm experimental farm of Wageningen University and Research Centre, Wageningen, the Netherlands (GPS coordinates, 51°59′18.0″ N; 5°39′40.1″ E). The soil of the experimental field plot, of 12 m × 5 m in size, was covered with black mulch foil, to avoid excessive weed growth during crop plant cultivation, and the entire plot was enclosed by a metal frame construction and covered with a bird net. The soil at the Unifarm field location was a sandy soil consisting of 86% sand, 9% silt, 1% clay and an organic matter content of 3.9% and pH of 5.9. Three subplots of 2 × 5 m were located within the experimental plot and boreholes were made using a soil bore of 6.5 cm in diameter to a maximal depth of 7 cm. Each subplot consisted of 12 rows of 15 boreholes and the distance between each borehole was 20 cm. Soil collected from the boreholes was used for the following soil treatments: (i) soil mixed with 3.5% fresh untreated cow manure (‘manure’ treatment), (ii) soil with fresh cow manure (3.5%) mixed with Log 7.15 strain 0611 CFU per g manure (M0611H), (iii) soil with fresh cow manure (3.5%) mixed with Log 5.60 strain 0611 CFU per g manure (M0611L) and (iv) soil with 0.4 mg mineral fertilizer per g dry soil (NPK) (Table 1). Upon soil treatment, mixed soils were placed back into the boreholes and one day later, indicated as 0 days after planting (DAP), planted with young lettuce (*Lactuca sativa* cultivar Lollo Rossa; HortiTops seeds, Heerenveen, The Netherlands) (one plot) or winter leek (*Allium porrum*, cultivar Pluston F1, BASF Nunhems, the Netherlands) plants (two plots, one for sampling in 2018 and further denoted as leek 2018 and the other for sampling after winter time in 2019 and further denoted as leek 2019). All plants were obtained from commercial providers of lettuce and leek plants and were pre-grown for 7–10 days in peat pellets. Each plant treatment consisted of four soil treatments, and in turn, each soil treatment consisted of 30 plants, resulting in a total of 120 plants per plant treatment. Four remaining rows were left open for another experiment, not related to this study.

### 2.3. Recovery of Escherichia coli 0611 from Manure, Soil and Root Samples

Transmission of *E. coli* strain 0611 via soil mixed with manure to growing plants was investigated. For manure, single samples were taken from untreated manure and from manure samples M0611H and M0611L, 3 h after mixing with strain 0611. For bulk soil, three subsamples per treatment were taken short before planting (0 DAP) from different boreholes in the subplots and from untreated soil between subplots, all at 7 cm depth. Rhizosphere soil, root and leaf samples from 120 plants (10 per soil and 40 per plant treatment) were randomly sampled from the field at 39 DAP for lettuce, at 90 DAP for leek 2018 and at 272 DAP for leek 2019 (Table 1).

Manure and bulk soil samples (5 g) were transferred to sterile 50 mL vials containing 9 mL 0.1% Sodium pyrophosphate (NaPPi) solution and 1 g gravel. Rhizosphere soils from lettuce and leek plants were obtained by vigorously shaking of plants with roots upon removal from soil. Soil still adhering to the roots was considered as rhizosphere soil. Root fragments of 5–7 cm were removed from the plants using sterile scalpel knives and subsamples from individual plants were transferred to sterile vials with NaPPi solution and gravel. Flasks with manure, soil or roots were shaken for 10 min at room temperature. The obtained manure and soil suspensions were either plated (50 µL) onto BECSA + Cf, for strain 0611 colony formation, or transferred to sterile microcentrifuge tubes (10 mL), spin down at 7000× *g* for 10 min at 4 °C, after which cell pellets were stored frozen at −70 °C for later soil DNA extraction, followed by amplicon sequencing, metagenome and TaqMan analyses. Roots (between 2–5 g) were collected from the rhizosphere soil suspensions using a 1 mm mesh sieve. Roots and leaves, between 2–5 g and cut from plants with sterile scalpel knives, were washed three times in sterile tap water and transferred to BioReba bags (Bioreba AG, Reinach, Switzerland) containing 5 mL Ringers solution and were crushed with a plastic hammer. The obtained root and leaf homogenates (50 µL) were then plated onto BECSA + Cf. All plates, containing tenfold diluted (in Ringer solution) or undiluted manure, soil or plant suspensions were incubated overnight at 37 °C. Upon incubation, purple colonies with morphologies identical to strain 0611 were enumerated. Strain 0611 densities calculated per sample were corrected for dry soil (bulk and rhizosphere soils) or fresh plant weights (roots and leaves), converted to Log_10_ (n + 1) (Log) values and expressed as Log strain 0611 CFU per g manure, dry soil or fresh plant.

DNA was extracted from frozen soil pellets of all bulk and rhizosphere soil samples, using the Qiagen MagAttract PowerSoil DNA Isolation Kit (Qiagen, Hilden, Germany). The manufacturer’s protocol was adapted for larger input in soil weight to up to one gram per sample. Volumes of all buffers were proportionally adapted in all downstream processing steps, including bead-beating and cell lysis. DNA washing steps were performed using the King Fisher platform and a double binding step was included to accommodate all available lysate per sample. Resulting DNA eluates were pooled per sample and stored at −20 °C. DNA concentration was measured with a Pico Green assay (Quant-IT Pico Green dsDNA Assay kit, Invitrogen) on a Tecan Infinite M200Pro plate reader (Tecan Group Ltd., Männedorf, Switzerland). DNA from manure was isolated using the QIAGEN QIAamp DNA Stool Mini kit, all according to the protocol provided by the manufacturer. For all samples, DNA was diluted to 4 ng per µL with elution buffer. Thus, normalized DNA extracts were used for all further molecular analyses.

Three TaqMan systems were designed, aimed to direct loci of the *bla*CTX-M-55, *qnr*S1 and *rec*A genes located on the strain 0611 genome (Appendix A), using the PrimerQuest TaqMan assay IDT design tool (Integrated DNA Technologies, Coralville, IA, USA). Primer and probe sequences were checked for specificity in the NCBI RefSeq genome database, using the NCBI Blastn algorithm. Calibration curves of the three TaqMan systems were made using a 399 bp synthetic DNA fragment containing complementary sequences to primers and probe of all three systems (IDT gBlock^TM^, Appendix A) and ten-fold dilutions of between 10^6^ to 1 copy of the gBlock molecule per µL were made in duplicate for each TaqMan system. Simplex TaqMan assays were performed in TaKaRa PCR mastermix (Takara Bio Europe SAS, Saint-Germain-en-Laye, France) containing 10 µM of each primer, 5 µM probe and 10 ng target DNA. Reactions were run in a Bio-Rad CFX384 thermocycler (Bio-Rad, Hercules, CA, USA) using an initial denaturation cycle at 95 °C for 2 min, followed by 40 cycles at 95 °C for 15 s, and at 60 °C for 1 min. DNA extracts of in total of 60 rhizosphere soil (four treatments of lettuce, leek 2018 and leek 2019, each with five replicates per treatment), nine bulk soil (soil with manure, M0611H or M0611L; three replicates per treatment) and three manure (manure, M0611H, M0611L) samples were screened for presence of strain 0611-specific DNA fragments with the three TaqMan systems, using the same DNA amplification conditions. Cycle threshold (Ct) values were converted to Ceq values by making use of the calibration curve corresponding to each TaqMan system. Per sample, the densities of strain 0611 were corrected for fresh manure or dry soil weights, converted to Log Ceq values and expressed as Log Ceq per g manure or dry soil.

### 2.4. Rhizosphere Soil Bacterial Community Analysis

Amplicon sequencing was performed on bulk and rhizosphere soil DNA extracts (three and five replicates per treatment, respectively) to establish the impact of soil treatments on bacterial soil community composition. Therefore, universal PCR primers E341F 5′-CCTACGGGNGGCWGCAG-3′ and 805R 5′-GACTACNVGGGTWTCTAATCC-3′ were applied to target the bacterial V3-V4 hypervariable region of the 16S rDNA gene [32,33]. Primers were synthetized with the universal Illumina MiSeq adapters attached to the primers (Integrated DNA Technologies, BVBA, Leuven, Belgium). PCR reactions were performed in 1 × Q5 Reaction Buffer, 200 μM dNTPs, 0.5 µM of each primer, 20 ng template DNA, 1 U Q5^®^ Hot Start High-Fidelity DNA Polymerase (New England Biolabs, MA, USA) and nuclease-free water to a final volume of 50 µL. PCRs were run at 98 °C for 30 s, for one cycle, followed by 15 cycles at 98 °C for 10 s, 55 °C for 30 s and at 72 °C for 30 s, with a final DNA elongation cycle at 72 °C for 2 min, using a Veriti Thermo Cycler (Thermo Fisher Scientific, Waltham, MA, USA). Four replicate PCRs were performed for each sample, in separate PCR runs, and replicates were pooled and stored at −20 °C. Amplicons were purified and a second PCR step was performed to add sample-specific barcodes to the amplified products. Libraries were prepared according to Illumina guidelines (Illumina, San Diego, CA, USA) and were paired-end sequenced at two times 250 cycles on an Illumina MiSeq platform, after which all reads were demultiplexed by ‘Next Generation Sequencing Facilities’ (Wageningen University and Research Centre, Wageningen, the Netherlands). All sequences were submitted to the NCBI database under the BioProject number PRJNA657959.

Pre-processing of the demultiplexed reads was performed in QIIME2 [34] v 2018.4. Raw sequences were denoised by run-specific quality control and filtering steps, including primer removal, merging of paired-end reads and chimera filtering in the DADA2 q2 plugin [35]. The resulting DADA2 output included a table with abundances of amplicon sequence variants (ASVs) [36] per sample. Low abundant ASVs (<0.0005%) were removed from the ASV table. Taxonomy was assigned to the ASVs with the Naive Bayes classifiers plugin in QIIME2 [37,38]. The 16S classifier was pretrained on extracted 16S V3–V4 region of the Silva 16S/18S database release 132 [39]. Amplicon sequence variants that remained unassigned at Kingdom level, or that were assigned to plant cell organelles, such as mitochondria and chloroplasts, were removed from the ASV table. Based on ASV abundances per sample, values for bacterial community diversity parameters ‘observed richness’ and ‘Shannon diversity’ in lettuce, leek 2018 and leek 2019 rhizosphere soils were calculated using the R package Phyloseq [40].

### 2.5. Screening for Strain 0611-Specific Reads in Manure, Bulk and Rhizosphere Soil Metagenomes

For metagenome screening, sheared DNA extracts from 32 samples, i.e., from manure (1), M0611H (1), untreated bulk soil, i.e., soil taken from the field before manure amendment (3) and from rhizosphere soil samples from lettuce (9), leek 2018 (9) and leek 2019 (9) under manure, M0611H and NPK treatments were paired-end sequenced (150 bp) using the NovaSeq 6000 platform (Illumina, San Diego, CA, USA). Obtained sequence reads that were shorter than 30 bp, as well as the ones containing ambiguous bases with an average score lower than 30, were discarded. The remaining reads were assembled using MEGAHIT software (version 1.1.3). Reads of individual libraries were mapped on the strain 0611 contigs using BBMap (version 38.22) software (default settings) [41] at 100% nucleotide identity setting. Number of identical reads per sample were normalized to the total number of mapped reads and expressed in number of reads per kilo base per million mapped reads (RPKM).

### 2.6. Statistical Analyses

Significance of differences between average Log strain 0611 CFU (*n* = 10), Log Ceq (*n* = 5), arcsin-converted RPKM (*n* = 5), observed richness (*n* = 5) and Shannon diversity (*n* = 5) values of rhizosphere soils under manure, M0611H, M0611L (not for arcsin-converted RPKM values), and NPK treatments were calculated using two-way ANOVA analyses (Genstat 19th Ed. Hemel Hempstead, UK). Comparison between average Log Ceq values from the three TaqMan systems with average Log CFU values from corresponding rhizosphere soil samples was performed using Student’s t-test. The effects of ‘manure’, ‘presence of *E. coli* strain 0611′ and ‘inoculum level of *E. coli* strain 0611′ as factors on ASV composition in lettuce, leek 2018 and leek 2019 rhizosphere soils under the four different soil treatments were calculated, using square root-transformed ASV counts, by canonical correspondence analysis (CCA), including a permutation test with 999 permutations, in R (https://www.R-project.org/ accessed on 10 September 2021). Calculation for the significance in differences in bacterial community composition in lettuce, leek 2018 and leek 2019 rhizosphere soils by ‘manure’ and ‘presence of strain 0611′ was done by, respectively, comparing ‘NPK’ with ‘manure’ and ‘M0611H’ with ‘manure’ treated samples. Therefore, a PERMANOVA test was performed on Bray distance matric values of the ASV datasets from these samples (999 permutations) using the R package vegan [42].

Comparison in Log-converted relative ASV values of the genera of *Clostridium sensu stricto (ss)*, *Paeniclostridium* and *Romboutsia* between four soil treatments in bulk soils and lettuce, leek 2018 and leek 2019 rhizosphere soils was performed by a Kruskall–Wallis test followed by a Dunn post-hoc test or by ANOVA followed by a Tuckey post-hoc test, depending on the distribution of the data and the homogeneity of the variance. For all statistical tests, differences were considered to be significant at levels of *p* ≤ 0.05, unless stated differently.

## 3. Results

### 3.1. Lettuce and Leek Plant Growth in the Field and Colonization of Field-Grown Plants by Strain 0611

Lettuce and leek 2019 plants were sampled at plant growth stages that are realistic for harvest in practice. At the moment of sampling in September 2018 (39 DAP), lettuce heads were still compact, and no flower stalks had developed yet. In the first sampling of leek plants in October 2018 (90 DAP), plant stems reached a diameter of approximately 1.5 cm, whereas in the second leek sampling, in May 2019 (272 DAP), plant stems reached a diameter of approximately 3 cm. *Escherichia coli* strain 0611, introduced at two density levels (Log 7.15 and Log 5.60 strain 0611 CFU per g manure) via manure into soil, was recovered as CFUs from lettuce rhizosphere soil and roots and from leek 2018 and leek 2019 rhizosphere soils (Figure 1). No strain 0611 CFUs were found in lettuce rhizosphere soil under manure and NPK treatments, which indicates that no background Cf-resistant *E. coli* populations were present in plant starting materials and in soil and manure before the start of the experiment. Initial strain 0611 CFU numbers at planting (0 DAP) were Log 4.15 (M0611L) and Log 5.70 (M0611H) per g dry soil (Table 1). Compared to these original soil inoculum densities, strain 0611 CFUs declined over time in the rhizosphere soil of lettuce to Log 1.40 (M0611L) and Log 4.11 per g dry soil (M0611H), in that of leek 2018 to Log 1.49 (M0611L) and Log 3.95 CFU per g dry soil (M0611H) and in that of leek 2019 to below detection (<Log 0.2 CFU per g dry soil; M0611L), and to Log 2.56 CFU per g dry soil (M0611H). Average strain 0611 CFU numbers in the lettuce rhizosphere soil was significantly higher in the M0611H than in the M0611L treatment, which indicates that final strain 0611 densities in rhizosphere soils and roots of both plant species were inoculum dose-dependent. In lettuce roots, average strain 0611 CFUs were only found in the M0611H treatment at the level of Log 2.17 CFU per g plant, and no strain 0611 CFUs were found under manure, M0611L and NPK treatments (detection level was Log 0.12 CFU per g fresh plant). In leek roots, strain 0611 CFUs were only found in the M0611H treatment in 2018 and 2019, respectively, at Log 1.72 and Log 0.34 per g plant and no strain 0611 CFUs were found in the other three treatments in both years. In lettuce leaves, one strain 0611 colony was found in one sample (out of 10) in the M0611H treatment and none in leek leaves in both years under any of the four applied soil treatments.

### 3.2. Impact of Plant and Soil Treatments on the Bacterial Community Composition in Bulk and Rhizosphere Soils

The impact of the four soil treatments on the lettuce and leek rhizosphere soil bacterial diversity, expressed in ASV richness and Shannon diversity, differed between treatments (Appendix A). No significant differences in average ASV richness values were found among the four soil treatments for lettuce (between 1559 and 2643) and for leek 2019 (between 2358 and 3340) rhizosphere soils. For leek 2018 rhizosphere soil, however, the average ASV richness was significantly higher for the M0611L (3647) than for the other three treatments (between 1892–2854). Across all plant treatments, NPK treatment resulted in significantly lower average ASV richness values (2221), than in the strain 0611-amended and unamended manure treatments (between 2695 and 2951), which in turn did not significantly differ from each other. Across soil treatments, significantly lower average ASV values were found in lettuce (2287) than in leek 2018 (2801) and leek 2019 (2932) rhizosphere soils, which in turn did not significantly differ from each other. Average Shannon diversity index values in lettuce rhizosphere soils was the same for treatments M0611H (6.99) and M0611L (6.91), and average values for both treatments were significantly higher than for the manure (6.62) and NPK treatments (6.44), and these two treatments did not significantly differ from each other. For leek 2018, the average Shannon diversity index values were significantly higher for the M016L treatment (7.51) than for the other three treatments (between 6.93 and 7.18), whereas no significant differences in average values were found between the four treatments in leek 2019 (between 6.86 and 7.21). Across all plant treatments, treatment with NPK resulted in significantly lower average Shannon diversity index values (6.83) than with all three manure treatments that did not significantly differ from each other (between 6.99 and 7.12). Across all soil treatments, significantly lower average Shannon diversity index values were found for lettuce (6.73) than for leek 2018 (7.19) and leek 2019 (7.08) and both leek treatments did not significantly differ from each other.

Clear differences in the bacterial composition, at normalized taxonomical phylum and genus levels, were present between treated bulk soil samples at the start of the field experiment and corresponding samples of lettuce, leek 2018 and leek 2019 rhizosphere soils (Appendix A). Differences between treatments were present for ASVs representing the genera of *Clostridium ss, Paeniclostridium* and *Romboutsia* (all Firmicutes) that were all three more abundant in manure, M016H and M016L, than in NPK-treated bulk soils at the start of the experiment (Appendix A). Significant differences between manure, M016H and M016L treatments on the one hand, and NPK on the other hand were also present in lettuce (for *Clostridium ss* and *Romboutsia*) and leek 2018 (all three ASV groups) rhizosphere soils, whereas differences were absent in leek 2019 rhizosphere soils.

Overall, in bulk and rhizosphere soil and manure samples, a total of six ASVs were found to be affiliated with the family of *Enterobacteriaceae*. Of these six ASVs, one was affiliated with the *Escherichia-Shigella* genus, to which strain 0611 belongs. *Escherichia-Shigella* amplicon reads were found in 20 bulk and rhizosphere soil and manure samples. The relative reads, expressed as the percentage reads per total reads of each sample, was 13.9% in M0611H, and this percentage was higher than in M0611L (0.15%) and manure (0.069%) (Table 2). When mixed through soil, the abundance of *Escherichia-Shigella* amplicons was again higher with M0611H (0.88%) than with M0611L (0.014%) or manure (0.0029%). *Escherichia-Shigella* ASVs were found in lettuce (0.034%) and leek 2018 (0.0064%) rhizosphere soils under M0611H treatment. No *Escherichia-Shigella* ASVs were found in both rhizosphere soils under M0611L, manure and NPK treatments, nor in the leek 2019 rhizosphere soils under all treatments.

Canonical correspondence analysis (CCA) on bacterial community composition of rhizosphere soil samples revealed separate clustering per treatment. Vectors in the biplots (Figure 2) indicate significant impacts of treatments on the bacterial community structures of lettuce, leek 2018 and leek 2019 rhizosphere soils. The first two axes in the CCA biplot explained 17.1% for lettuce, 14.4% for leek 2018 and 14.8% for leek 2019, of all variation between samples. Treatment with NPK always led to segregation from manure, M0611H and M0611L samples, whereas treatment with manure occasionally led to segregation from M0611H and M0611L samples in cases of lettuce and leek 2019 treatments. Soil treatments M0611L and M0611H led in two cases (leek 2018 and leek 2019) to segregation of samples, whereas the leek 2018 samples resembled each other more under manure and M0611H than the ones under M0611L and NPK treatments. The impact of ‘manure’ by comparing samples under NPK and manure treatments and the impact of ‘presence of strain 0611′, by comparing M0611H and manure treatments with each other by performing PERMANOVA on Bray-Curtis distance matrix values revealed significant differences (*p* values between 0.008 and 0.033 and R^2^ values between 0.15 and 0.24, respectively) in all three plant treatments.

### 3.3. Rhizosphere Soil Metagenome Screening on Presence of Strain 0611-Specific Reads

Whole genome sequencing of strain 0611 resulted in 125 contigs of 240 bp to 395 kb in length. The estimated genome size was 4.71 Mb (N50 value of 165 kb). One contig (68), of 3.5 kb in size, possessed the *rep*A gene of an IncY plasmid and most likely this contig originates from a plasmid (Table 3). Six antibiotic resistance genes were found present on the strain 0611 genome located on four different contigs. Contig 9 (175 kb in size) possessed three antibiotic resistance genes, including the *bla*CTX-M-55 gene, for which the strain was originally selected. Other resistance genes on contig 9 conferred resistances against ciprofloxacin and sulfonamide. On contig 52 (11.6 kb), a tetracycline, on contig 65 (4.1 kb) a gentamycin, and on contig 73 (2.0 kb) a trimethroprim resistance gene was present. The *sul*2 gene located on contig 9 showed close resemblance to *sul*2 genes found in *Acinetobacter baumannii*, indicating that this gene was acquired via HGT. No virulence genes typical for STECs, i.e., *eae*, *hly*A, *stx*1 and *stx*2 [43], were found present on the genome of strain 0611, although contig 9 harbored an IS91-family transposase gene. The strain belonged to sequence type (ST) 196, based on in silico sequence alignments with seven loci (*adk*, *fum*C, *gyr*B, *icd*, *mdh*, *pur*A, *rec*A), which are located on *E. coli* chromosomes retrieved from public DNA sequence depositories.

Reads identical to contigs 9, 52, 65, 68 and 73 of *E. coli* strain 0611 were found in metagenome libraries from manure (untreated and M0611H), untreated soil, and lettuce and leek rhizosphere soil samples (manure, M0611H and NPK treatments). Obviously, the highest relative read numbers, expressed in RPKM, were found in the M0611H samples, ranging between 6.39 and 5.25 for the five contigs. However, in untreated manure and untreated soil samples, relative read values for the five contigs were occasionally found present between, respectively, 0.05 and 0 and between 0.03 and 0. Reads identical to contigs 9, 52, 65, 68 and 73 were found in, respectively, 26, 24, 1, 6 and 26 of 27 analyzed lettuce and leek rhizosphere soil metagenome samples and relative read values are shown in Figure 3 for contigs 9, 52, 68 and 73, whereas the ones of contig 65 were omitted from the figure due to its low prevalence. A significantly higher average contig 9 RPKM value was found in the lettuce rhizosphere soil metagenome under the M0611H treatment (0.013) than in the other eight metagenomes (between 0.0013 and 0.0040) (Figure 3). No significant differences in the average RPKM values for contigs 52, 68 and 73 were found in all three rhizosphere soil metagenomes over the different treatments. Reads identical to contigs 9 and 73 were found in all three rhizosphere soil metagenomes of plants grown under NPK treatment, and occasionally, reads identical to contigs 9, 52, 68 and 73 were found in all three rhizosphere soil metagenomes of plants grown under manure treatment. The overall picture of relative read numbers of contig 9 and 52 over lettuce and leek rhizosphere soil metagenomes under the different treatments resembled that of strain 0611 CFU numbers (Figure 1), which was, however, not the case for contigs 68 and 73.

TaqMan systems, designed to target the *bla*CTX-M-55 and *qnr*S1 genes, located on contig 9, and the *rec*A gene of strain 0611, gave strong positive signals to DNA extracts from M0611L (Ct values between 26.7–29. 3) and M0611H (Ct values between 20.9–22.7) samples. Upon conversion of Ct values into Ceqs, by making use of a gBlock calibration curve, it revealed that in most cases the obtained average values did not significantly deviate from average Log CFU values from the same sample. For M0611L, the average Log Ceq values per g manure were 5.06 for *bla*CTX-M-55, 4.95 for *qnr*S1, 5.01 for *rec*A, whereas the corresponding CFU value was Log 5.60 per g manure, and for M0611H, values were 7.17 for *bla*CTX-M-55, 7.13 for *qnr*S1, 7.07 for *rec*A, whereas the corresponding CFU value was Log 7.15 per g manure. Cycle threshold values made with the three TaqMan systems on all bulk and rhizosphere soil DNA samples varied between 21.5–40 and Ct values of 38 and higher were considered as ‘not detectable’. Cell equivalent numbers, made with the three TaqMan systems and applied to DNA extracts made from bulk soil samples under M0611H and M0611L treatments also did not significantly deviate from corresponding CFU values. For soil with M0611L, average Log Ceq values per g dry soil were 3.66 for *bla*CTX-M-55, 3.48 for *qnr*S1, 3.81 for *rec*A, whereas by colony growth, the value was Log 4.15 per g dry soil, and for soil with M0611H, values were 6.02 for *bla*CTX-M-55, 5.97 for *qnr*S1 and 5.86 for *rec*A, whereas by colony growth the corresponding value was Log 5.70 per g dry soil. Positive signals with all three TaqMan systems were also obtained in reactions with DNA extracts from lettuce and leek rhizosphere soils (Figure 4). In DNA extracts from lettuce rhizosphere soil under M0611L treatment, positive signals were obtained in two (*bla*CTX-M-55, *qnr*S1) and five (*rec*A) of five analyzed samples. In extracts made from lettuce rhizosphere soil under M0611H treatment, all five replicates were positive for the three TaqMan systems. Significantly higher Log Ceq values per g dry soil were found in lettuce rhizosphere soil under M0611H treatment than in the same rhizosphere soil under M0611L treatment, with all three TaqMan systems. Furthermore, the TaqMan systems directing the *bla*CTX-M-55 and *qnr*S1 genes did not produce positive signals in lettuce rhizosphere soil under manure and NPK treatments, whereas the *rec*A TaqMan system did. In the leek 2018 rhizosphere soil, Log Ceq values per g dry soil were significantly higher when plants were grown under M0611H than under M0611L treatment for all three TaqMan systems. No positive signal was found in the leek 2018 rhizosphere soil under M0611L treatment with the TaqMan system directing the *bla*CTX-M-55 gene. In DNA extracts from leek 2019 rhizosphere soil, positive signals were found in the M0611H treatment with all three TaqMan systems and values were higher than when derived from the other three soil treatments. In leek 2019 rhizosphere soil samples under M0611L treatment, no positive signal with the TaqMan systems directing *bla*CTX-M-55 and *qnr*S1 genes were found, whereas with the *rec*A-targeting TaqMan system, positive signals were obtained in rhizosphere soils under all treatments. The Log Ceq values of the three TaqMan systems from lettuce and leek 2018 rhizosphere soils under M0611H treatment did not significantly differ from corresponding CFU values. For lettuce, Log Ceq values per g dry soil were 4.76 for *bla*CTX-M-55, 4.63 for *qnr*S1, 4.41 for *rec*A, whereas by colony growth, the corresponding CFU value was Log 4.11 per g dry soil and for leek 2018, Log Ceq values per g dry soil were 4.20 for *bla*CTX-M-55, 3.97 for *qnr*S1, 4.00 for *rec*A, whereas the corresponding CFU value was Log 3.95 per g dry soil. However, significant differences between Log *bla*CTX-M-55 (3.36) and *rec*A (3.21) Ceq and Log CFU (1.96) values per g dry soil were present in leek 2019 rhizosphere soil, resulting in Δ values (Log Ceq-CFU) of, respectively, 1.40 and 1.25. The Ceq value per g dry soil for the *qnr*S1 TaqMan system in the same sample was 2.73 and this value did not significantly differ for the corresponding Log CFU value (Log Ceq-CFU value of 0.77).

## 4. Discussion

*Escherichia coli* strain 0611 was transmitted from manure, via soil to the plant root zone, where, for leek plants, it survived during winter time up to 272 DAP. However, the fact that only one strain 0611 colony was found in lettuce, and none in leek leaves indicates that manure is not the principal transmission route of *E. coli* to the edible parts of both plant species in this study. This is an important finding, because contamination of fresh vegetables with human pathogens is often linked to the use of fresh manure at plant production [5,6,7,20]. The fact that leave contamination with *E. coli* strain 0611 in both plant species was virtually absent indicates that no secondary spread took place under field conditions, but it also indicates that strain 0611 did not systemically spread, at least not in massive quantities, from roots to the above-soil compartments of lettuce and leek plants. This confirms the results from the study performed with *E. coli* O104:H4, introduced via pea seeds into the root zone of pea plants grown in the open field, where *E. coli* only sporadically spread from roots to shoot tissue [15]. Accumulation of human pathogens in the root zones of growing plants can still be a risk when contaminated soil particles come into contact with leaves, which is often the case for plants growing close to the soil surface, such as lettuce plants. Soil splashing via heavy rainfall or overhead irrigation is realistic and might lead to secondary spread of human pathogens to the edible parts of plants, and therefore, soil contamination can be considered as a potential hazard to human health.

Long-term survival of *E. coli* strains, introduced via manure into bulk soil [21,25,26] and in maize and cabbage rhizosphere soils [7,20], has been demonstrated before. From that perspective, *E. coli* persist in bulk and rhizosphere soils and can be considered as an intrinsic member of soil microbial communities as previously proposed [23,44]. Cells of *E. coli* strain 0611 seem to be attracted to lettuce and leek root zones, where they accumulate and adhere to root surfaces. From these observations, it can be concluded that *E. coli* strain 0611 was able to colonize root zones of both plant species. Cells of the same strain were also shown to colonize roots of pea plants [15]. The fact that *E. coli* is able to colonize root zones of three distinct plant species would indicate that this species can possibly colonize a taxonomically wide variety of plant species. Small differences in density levels in lettuce and leek rhizosphere soils and roots may be the result of the later harvest of leek 2018 (90 DAP) than of lettuce plants (39 DAP), but also may reflect eventual differences in root exudation patterns.

The introduced strain was recoverable from rhizosphere soil by selective medium plating and by molecular analysis for presence of *bla*CTX-M-55, *qnr*S1 and *rec*A genes. The TaqMan system directing the *rec*A gene of strain 0611 gave positive signals in all rhizosphere samples, even in samples where no signals were measured with TaqMan systems directing the *bla*CTX-M-55 and *qnr*S genes. The *rec*A gene is a housekeeping gene and expectedly more conserved among a wider group of *E. coli* strains than strain 0611 alone. This would explain the higher prevalence of this gene in rhizosphere soils under different treatments. Additionally, it would indicate that ‘indigenous’ *E. coli* strains could be present in the different rhizosphere soils, as signals were also found in NPK-treated rhizosphere soils, where there was no input from animal manure. This observation is strengthened by the fact that *Escherichia-Shigella* ASVs were found in bulk soil amended with NPK. As already suggested in Jang et al. [44], *E. coli* is a species that is intrinsically present in bulk and rhizosphere soils and absence of positive TaqMan signals for the *bla*CTX-M-55 and *qnr*S genes indicate that target sites for these systems were absent in soil-indigenous *E. coli* populations.

Measured CFU and Ceq values in manure, manured-bulk soil and lettuce and leek 2018 rhizosphere soils were statistically indistinguishable from each other. This demonstrates that strain 0611 cells were detectable as culturable cells. However, for leek 2019 rhizosphere soil under M0611H treatment, when cells persisted in rhizosphere soil over a period of 272 DAP, Ceq values where higher than CFU values. Based on Δ Log Ceq-CFU values, 6–25 times higher Ceq than CFU numbers were measured in these samples, indicating that measured signals by the TaqMan approach may have originated from lysed or dead cells or from cells that were still intact but in a viable-but-non-culturable (VBNC) state. In soil inoculation experiments conducted with *Pseudomonas fluorescens*, it was demonstrated that resistance to different stress factors increased upon long-term residence in bulk and rhizosphere soils and induced the VBNC state of the introduced strain [45]. Therefore, it is plausible that a fraction of the introduced *E. coli* strain 0611 population persisted in rhizosphere soil in a soil-adapted form, resisting stressful circumstances commonly present in soil environments [12,45].

The presence of strain 0611 in lettuce and leek rhizosphere soils under different manure treatments was confirmed by measurements of *Escherichia-Shigella* ASVs and metagenome reads identical to strain 0611 contigs 9 and 52. Antibiotic resistance genes located on contig 9 of the strain 0611 genome were most specific for this strain, which was confirmed both by targeted detection using the *bla*CTX-M-55 and *qnr*S-directed TaqMan systems and by selection of contig 9 reads from the rhizosphere soil metagenomes. Based on the significant higher number of contig 9 reads in the lettuce rhizosphere soil under M0611H treatment than under the other three treatments, a reliable estimation of Log 4.11 CFUs can be made for strain 0611 detection in a shotgun metagenome library. This indicates that at this level, the introduced *E. coli* strain was reliably detectable in rhizosphere soil metagenomes. Untargeted approaches offer potential opportunities to screen for a wide spectrum of human pathogens in field production soils. In this study, we demonstrate that only specific parts of the genome of the introduced *E. coli* strain can be reliably detected via a metagenomic approach, whereas other parts cannot, because of too much homology with genomes of other *E. coli*-like bacteria. This is an important message for application of untargeted approaches in screening for presence of human pathogens in field production soils, such as metagenomics, namely that criteria for the selection of metagenomic reads must be well defined beforehand.

Amendment of manure and strain 0611 to lettuce and both leek rhizosphere soils has separate impacts on bacterial community composition. This implies that manure amendment to soil has led to a change in the bacterial community composition in rhizosphere soils. Bacterial community changes in rhizosphere soils will impact the ecological behavior of *E. coli* near growing plants, as demonstrated before [29,30]. The observed bacterial community shifts and increased bacterial diversity in lettuce and leek 2018 rhizosphere soils under M061H versus manure treatments cannot be related to the mere presence of strain 0611 in both rhizospheres only. Namely, in leek 2019 rhizosphere soil, there was still a significant impact of ‘strain 0611 amendment’ in spite of the fact that observed CFU and Ceq numbers were low, even under M0611H treatment, and no ASVs specific to *Escherichia-Shigella* were found in these samples. Amendment of strain 0611 to rhizosphere soil, thus, impacted residing bacterial communities, resulting in a longer lasting effect on the bacterial community composition in rhizosphere soil. Obviously, the three genera of *Clostridium ss, Paeniclostridium* and *Romboutsia* were transferred from the cow gut intestinal track system, via manure to the arable production system. This is an interesting observation that, besides *E. coli*, other microbial groups can also be transferred from animals via their manure to soils. *Clostridium ss, Paeniclostridium* and *Romboutsia* are anaerobic genera and commonly associated with manure and intestinal track systems of warm-blooded animals [46,47]. Some, but not all, species belonging to these three genera have the capacity to sporulate, which could explain their survival in rhizosphere soil. However, soil residence of these genera appears to be rather short as differences in relative abundances between all manure treatments versus NPK were no longer significantly different in 2019. The plant root zone can be considered as an alternative habitat for *E. coli*, which would explain the complete life cycle of *E. coli* as an intestinal track colonizer of grazing herbivores that are transmitted via fecal deposits and rhizosphere soil of pasture plant cover to other herds of grazers, as was proposed in a previous study [23].

## 5. Conclusions

Manure is an important transmission route for *E. coli* strain 0611 to plant root zones. However, manure application with strain 0611 did not result in contamination of the above-soil, and edible, parts of both plants species, lettuce and leek. This contradicts what is generally stated, i.e., that the application of manure leads to contamination of freshly consumed vegetables. The fact that strain 0611 accumulates over a period of 272 days in leek root zones still indicates that secondary transmission via soil splashing to edible plant parts is possible. Therefore, paying attention to soil contamination in vegetable production is important.

## Figures and Tables

**Figure 1 microorganisms-09-02289-f001:**
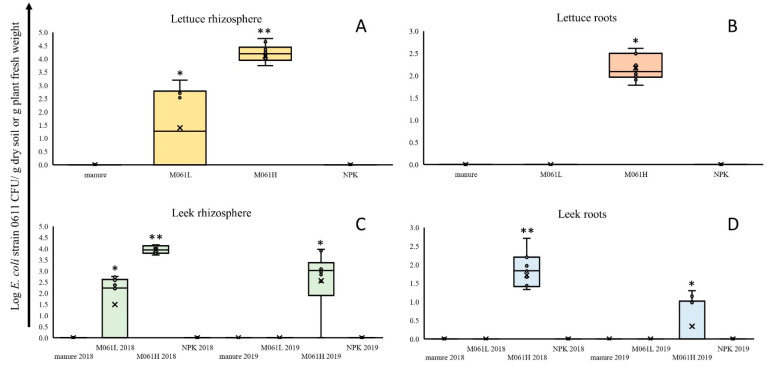
Colonization, as determined by *E. coli* strain 0611-specific colony formation, of rhizosphere soils (**A**,**C**) and roots (**B**,**D**) of lettuce plants sampled at 39 DAP (**A**,**B**) and of leek plants sampled at 90 (2018) and 272 DAP (2019) (**C**,**D**), grown in soils under four different treatment regimens (manure, M0611H, M0611L, NPK). Bars marked with * and ** indicate significant difference (*p* ≤ 0.05) in average Log CFU values between soil treatments, whereby ** > *.

**Figure 2 microorganisms-09-02289-f002:**
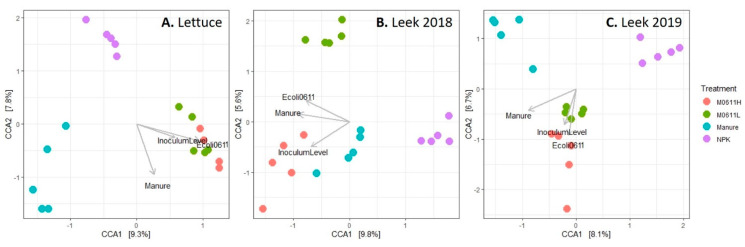
Canonical correspondence analysis ordination biplots on bacterial community composition of lettuce (**A**), leek 2018 (**B**) and leek 2019 (**C**) rhizosphere soil samples under four treatment regimens. Direction of vectors for treatments of ‘Manure’ (untreated manure, M0611H, M0611L) and NPK, ‘Inoculum Level’ (M0611H versus M0611L) and ‘*E. coli* 0611’ (presence of strain 0611 in M0611H and M0611L versus absence in untreated manure and NPK treatments) are indicated by arrows in the biplot. The impact of ‘Manure’ on bacterial community composition in lettuce, leek 2018 and leek 2019 rhizosphere soils was significant at levels of, respectively, 0.012, 0.011 and 0.013 (PERMANOVA, R^2^ values, respectively, 0.24, 0.18, 0.24) and of ‘*E. coli* 0611’ was, respectively, 0.008, 0.033, 0.025 (PERMANOVA, R^2^ values, respectively, 0.24, 0.17, 0.15).

**Figure 3 microorganisms-09-02289-f003:**
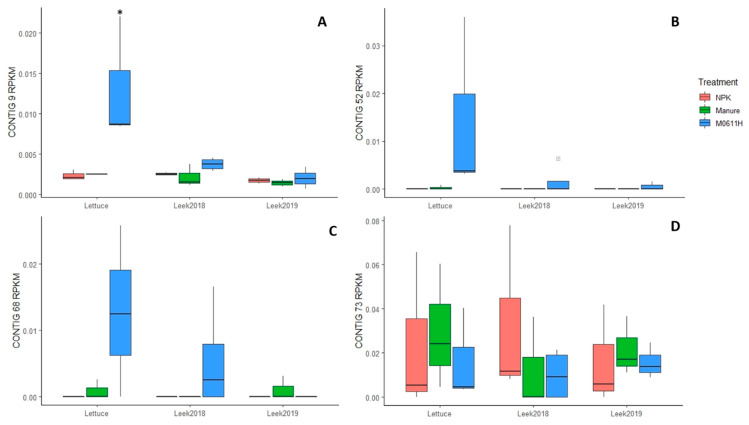
Normalized numbers of *E. coli* strain 0611 genomic reads, identical to contigs 9 (**A**), 52 (**B**), 68 (**C**) and 73 (**D**), in metagenomes of lettuce, leek 2018 and leek 2019 rhizosphere soils under three soil treatment regimens (manure, M0611H, NPK) and expressed in RPKM values. Average RPKM value marked with * indicate significant difference (*p* = 0.001) from other values of the same contig.

**Figure 4 microorganisms-09-02289-f004:**
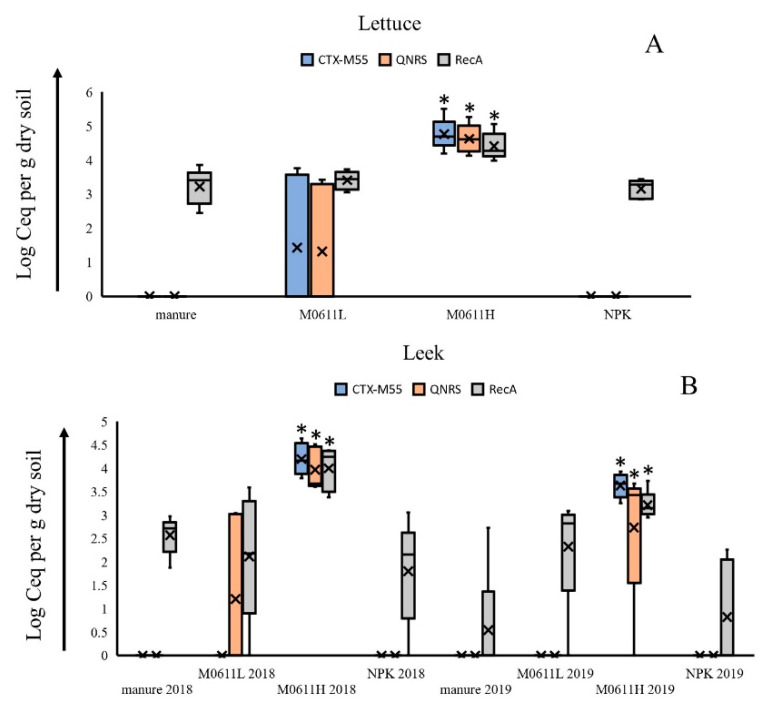
Molecular quantification of *bla* CTX-M-55, *qnr*S1 and *rec*A loci located on the strain 0611 genome and expressed in Ceq values in lettuce (**A**) and leek (**B**) rhizosphere soils under four treatment regimens, as determined by TaqMan. Average values marked with * indicate significant difference (*p* ≤ 0.05) from values of the other soil treatments determined with the same TaqMan system.

**Table 1 microorganisms-09-02289-t001:** Soil and plant treatments applied in field experimentation.

Soil/Plant Treatment	Description
Soil with manure (Manure)	Soil mixed with non-treated fresh cow manure (3.5% weight basis).
Soil with manure and strain 0611 high (M0611H)	Soil mixed with 3.5% fresh cow manure containing Log 7.15 strain 0611 CFU/g manure. Calculated density of strain 0611 in soil was Log 5.70 CFU/g dry soil, which corresponds to Log 8.07 CFU per plant.
Soil with manure and strain 0611 low (M0611L)	Soil mixed with 3.5% fresh cow manure containing Log 5.60 strain 0611 CFU/g manure. Calculated density of strain 0611 in soil was Log 4.15 CFU/g dry soil, which corresponds to Log 6.52 CFU per plant
Soil with NPK (NPK)	Soil mixed with NPK fertilizer to 0.4 mg per g dry soil, which corresponds to 100 mg per plant.
Lettuce	Four soil treatments, 30 plants per treatment. Plants were sampled at 39 days after planting (DAP).
Leek 2018	Four soil treatments, 30 plants per treatment. Plants were sampled at 90 DAP.
Leek 2019	Four soil treatments, 30 plants per treatment. Plants were sampled at 272 DAP.

**Table 2 microorganisms-09-02289-t002:** Normalized number of *Escherichia-Shigella* amplicon reads in manure, soil and lettuce and leek 2018 rhizosphere soil samples.

Sample Type	% *Escherichia-Shigella* Amplicon Reads of Total Reads per Sample (Total Number of Samples with Reads; Total Number of Samples)
Manure (non-treated)	0.069 (1; 1)
M0611H	13.9 (1; 1)
M0611L	0.15 (1; 1)
Soil with manure	0.0029 (3; 3)
Soil with M0611H	0.88 (3; 3)
Soil with M0611L	0.014 (3; 3)
Soil with NPK	0.0027 (1; 3)
Lettuce rhizosphere soil with M0611H	0.034 (4; 5)
Leek 2018 rhizosphere soil with M0611H	0.0064 (3; 5)

**Table 3 microorganisms-09-02289-t003:** DNA contigs with antibiotic resistance or plasmid backbone genes assembled by whole genome sequencing of *E. coli* strain 0611.

Contig	Resistance or Plasmid Type	Gene	Accession	Total Length (bp)	Identity (%)
9	Ampicillin, Ciprofloxacin Sulfonamide	*bla*CTX-M-55 *qnr*S1 *sul*2	GQ456159, AB187515, AY034138	876 657 816	100 100 100
52	Tetracycline	*tet*(A)	AF534183	1275	97.8
65	Gentamicin	*aac*(3)-IId	EU022314	861	99.9
68	plasmid	*rep*A (IncY plasmid)	K02380	765	100
73	Trimethoprim	*dfr*A14	DQ388123	483	99.6

## Data Availability

Not applicable.

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
