# Peer review of "Transmission of Escherichia coli from Manure to Root Zones of Field-Grown Lettuce and Leek Plants"

_microorganisms, 2021, doi:10.3390/microorganisms9112289_

Round 1

Reviewer 1 Report

The manuscript reports the capability of E.coli strain 0611 to contaminate the soil when seeded through manure and to persist in soil and in vegetables.

Although the data are interesting, unfortunately, the manuscript lacks substantial novelty (the persistence of E.coli in soils and vegetables is already known) and reports many analyses poorly pertinent, considering the aims of the paper. Two among others, it is difficult to understand why the authors analyzed the soil and rhizosphere microbial communities using NGS techniques to detect known sequences from a known (experimentally employed) strain, as well as the impact on natural microbial communities.

Other comments are included in the attached PDF file.

Author Response

We thank reviewer 1 for helpful comments. The abstract, part of the introduction and discussion sections were reorganized to make the objective of the study more clear. Moreover, part of the text in introduction and discussion section on horizontal gene transfer, including six references, were removed. We believe that the manuscript is more focused now. Concerning novelty of the experimental work, we believe it is, because not so many studies were performed on transmission of human pathogens or their proxies from manure to field-grown crops. Also the outcome is important because from the field study it is clear that the introduced E. coli strain is not colonizing the above-soil parts of the plants, at least not into a large extent. Colonization of roots and adjacent soil is colonized by the E. coli strain and that seems to be a general phenomenon in many plant species. Further, the experiment was carried out using a polyphasic detection approach, including cultivation-dependent and independent technologies. In plant sciences, different techniques are commonly applied to determine the ecological behavior of (introduced) microbes. Using the combination of techniques, we were able to show the critical points for every detection technique. This allows us to give a better overview on the ecological behavior of the introduced E. coli strain, and to E. coli in general, in a plant-soil system, realistic for agricultural circumstances. Metagenomic techniques is included in the analyses, because 1) it confirms targeted detection by the TaqMan approach, and 2) it may become an important technique to screen for human pathogens in production soils in the near future. We better emphasized these aspects in the latest version of our manuscript.

Other comments in the PDF file:

  • Change Latin names unto italics throughout the text
    • This was done, although lost to transition to the MDPI format.
  • Rephrase part of the abstract
    • Abstract is adapted
  • Epithilium into Epithelial
    • This is done now
  • Remove purpose of the experiments to results section
    • The changed wording in the text
  • Use capital letter for Lettuce in heading 3.1
    • This is done now
  • It is not clear the pertinence of such analysis with the topic and aims of the paper (in relation to section 3.2)
    • See response above. Multiple techniques are required in plant (environmental) science to describe the ecological behavior of microbes.
  • A PCR primer set is expected to be more sensitive than metagenomic analysis for specific (known) sequences detection. Moreover, strain re-isolation could have been much more convincing and informative about the actual presence of viable cells. Why did authors perform NGS analysis?
    • A primer (with probe) set is indeed more specific and very reliable when the exact target is known. This is the case with the introduced strain. However, it is the intention in later studies to describe the ecology of food-borne pathogens more in general and then the exact targets are unknown. This is the reason that we applied metagenomics to determine if the outcome via metagenomics will lead to the same conclusion as with the TaqMan approach, or not. We rewrote part of the discussion section to emphasize the importance of multiple detection techniques.

Reviewer 2 Report

Reviewers' comments:

Manuscript number: microorganisms-1401014

Title: Transmission of Escherichia coli from manure to root zones of field-grown lettuce and leek plants.

Comments: 

The manuscript reported on Transmission of Escherichia coli from manure to root zones of field-grown lettuce and leek plants. The manuscript needs a detailed editing. It cannot be recommended for publication in the present form. I hope the following points would be helpful for the authors.

- In the Abstract: the authors need to improve with more specific short results and conclusions.

- Escherichia coli….to….. Escherichia coli.

- The introduction section should be improved.

- E. coli…to… E. coli.

- 2.5. Screening for strain 0611 reads in manure, bulk and rhizosphere soil metagenomes – should be improve.

- Figure 1 – not clear make clear.

- Figures 2 and 3 – not clear make clear.

- 4. Discussion – should be improve.

- Several faults: are added or missing spaces between words: see manuscript file.

- Conclusion should be concise.

- References: there are recent references in 2021 treating the same subject, you can use and make all references in same format for volume number, page numbers and journal name, because it is difficult to searching and reading.

So that I recommended this manuscript to major revision and for future process.

Author Response

We thank reviewer 2 for helpful comments. We checked the text for spelling and editorial mistakes. However, editing was correct in the original WORD document, but was lost due to the shift to the MDPI format. We contacted the editorial office about this.

More into detail; we adapted the abstract and made discussion and conclusion sections more concise. As mentioned before, the Latin species names were correct in italic in the WORD document, but lost during transition to the MDPI format. The same for the references, they were correct in the original document, but lost in the MDPI format.

Concerning chapter 2.5, we made some changes in the text, but we prefer to keep the wording as it explains adequately the metagenome approach in our study. All legends of the figures were adapted and figures 2 and 3 were changed. However, we believe that all figures are relevant, but we will submit higher quality figures to the editorial office upon preparation of the final version of the manuscript. 

Reviewer 3 Report

Zdrojový text

                    973 / 5000    

Výsledky překladu

The presented manuscript addresses the current issue of environmental pollution by human pathogens. The problem is promising in relation to the quality and safety of food, but also in relation to plant health. Monitoring the effect of E. coli on the plant rhizosphere is therefore very important. The manuscript is written relatively carefully, however, I recommend reworking the abstract, in which the objectives of the work are given in great detail, but the description of the results is relatively short. Similarly, I probably would not mention the notion that many publications deal with this. This phrase reduces the topicality of this research. I would explain this in a similar way in the introduction. The methodology is sufficiently described. The results are well covered. However, the quality of the graphs is not sufficient, they are harder to read and lower quality. I recommend reworking. The discussion is rather descriptive, but there is a complete lack of reflection on why this is the case. The summary is fine and based on the results. References are well chosen.

Author Response

We thank reviewer 3 for the supporting comments to our study. Based on comments, we rephrased the abstract and made objections and results of the study more clear. We indeed removed the phrase on the many publications in the text and reworded the sentence. Further, we reorganized introduction and discussion sections, also based on comments of the other reviewers. The quality of the figures will be improved upon final submission as already discussed with the editorial office. We changed the lay-out of Figures 2 and 3 and modified text in all legends of the figures.

Reviewer 4 Report

The manuscript submitted by Overbeek et al., entitled “Transmission of Escherichia coli from manure to root zones of field-grown lettuce and leek plan” is well designed and conducted the experiments appropriately. The study aims to establish the fate and bacterial community impact of E. coli strain 0611, equipped with resistances to beta-lactam and five other antibiotics, and introduced via manure into soil planted to lettuce and leek. There are some minor modifications required in the present manuscript. The present form is not ready for publication.

  1. Use the recommended format to write the scientific name of organisms.
  2. There are several grammatical errors, which need a change in the flow and sentence construction.
  3. Certainly, in figure 3, I found the rate of significance indications are missing? Is it non-significant or missed is by any chance?

Author Response

We thank reviewer 4 for the supporting comments to the manuscript. We recognized that the format of our original document in WORD was lost during transition to the MDPI format and we contacted the editorial office about this. We checked phrases, wording and sentences throughout the text and made modification where necessary. Further, we add the level of significance in the legend of Figure 3.

Round 2

Reviewer 1 Report

The revised manuscript shows some improvement, I judge it interesting, but the main flaws remain, including the lack of substantial novelty and the experimental approach. Moreover, two points appear to be erroneous/misleading:

-L19-20 in the abstract: you should refer to reads, not to CFU/g, as NGS is not so quantitative.

-L103-108: if the aim was that, the experimental design should have been different, with the inoculation of the experimental strain (with and without manure) in plants with known (previously assessed) microbial assemblage.

As for your last reply to my comments: I have to judge the pertinence of the experimental approach with the aims of the current manuscript (where a known strain -with known molecular targets- was employed), not of later studies you are planning to do. Thus, the use of metagenomic approach, though very interesting, appears poorly appropriate (but fully fitting with the scopes you refer to for future experiments).